# Potential Beneficial Role of Nitric Oxide in SARS-CoV-2 Infection: Beyond Spike-Binding Inhibition

**DOI:** 10.3390/antiox13111301

**Published:** 2024-10-26

**Authors:** Sergio Sánchez-García, Antonio Castrillo, Lisardo Boscá, Patricia Prieto

**Affiliations:** 1Instituto de Investigaciones Biomédicas Sols-Morreale, CSIC-UAM, Arturo Duperier 4, 28029 Madrid, Spain; acastrillo@iib.uam.es; 2Unidad de Biomedicina (Unidad Asociada al CSIC), Universidad de Las Palmas de Gran Canaria, 35016 Las Palmas, Spain; 3Centro de Investigación Biomédica en Red de Enfermedades Cardiovasculares (CIBERCV), Av. Monforte de Lemos 3-5, P-11, 28029 Madrid, Spain; 4Departamento de Farmacología, Farmacognosia y Botánica, Facultad de Farmacia, Universidad Complutense de Madrid, Plaza Ramón y Cajal, 28040 Madrid, Spain

**Keywords:** nitric oxide, ACE2, spike protein, NO donors, COVID-19, infection

## Abstract

SARS-CoV-2, the causative virus for the COVID-19 disease, uses its spike glycoprotein to bind to human ACE2 as a first step for viral entry into the cell. For this reason, great efforts have been made to find mechanisms that disrupt this interaction, avoiding the infection. Nitric oxide (NO) is a soluble endogenous gas with known antiviral and immunomodulatory properties. In this study, we aimed to test whether NO could inhibit the binding of the viral spike to ACE2 in human cells and its effects on ACE2 enzymatic activity. Our results show that ACE2 activity was decreased by the NO donors DETA-NONOate and GSNO and by the NO byproduct peroxynitrite. Furthermore, we found that DETA-NONOate could break the spike–ACE2 interaction using the spike from two different variants (Alpha and Gamma) and in two different human cell types. Moreover, the same result was obtained when using NO-producing murine macrophages, while no significant changes were observed in ACE2 expression or distribution within the cell. These results support that it is worth considering NO as a therapeutic agent for COVID-19, as previous reports have suggested.

## 1. Introduction

In December 2019, the expansion of Severe Acute Respiratory Syndrome Coronavirus 2 (SARS-CoV-2) started in China, causing an infection that received the name of coronavirus disease 2019 (COVID-19) and that ended up triggering a pandemic in 2020 [1]. Although this disease is mild to asymptomatic in most patients, some people develop complications such as bilateral pneumonia or acute distress respiratory syndrome (ARDS), and in the worst scenario, COVID-19 results in the death of the patient [2].

SARS-CoV-2 uses its homotrimeric spike glycoprotein to bind to angiotensin-converting enzyme 2 (ACE2), which is the main host cell receptor [3]. The S1 subunit of the spike protein contains the receptor-binding domain (RBD), allowing interaction with the host ACE2. Consequently, refolding of the S2 domain drives the fusion of viral and host membranes [4]. As the S1 subunit is the peptide that recognizes the host, it has accumulated a great number of mutations that have enhanced viral infectivity [5]. Some of these mutations were the main cause of the different waves of infection and prompted the WHO to designate different variants of SARS-CoV-2, which are named by Greek letters, with the Alpha variant being the first one to be described [6]. As ACE2 acts as an entry route to several viruses such as SARS-CoV-2, levels of ACE2 expression in humans can be crucial for the success of the virus and to determine the severity of the disease [7,8].

At the cellular level, ACE2 is present in the plasma membrane as an integral protein, but it is also secreted to circulate in its soluble form [9]. Endogenously, ACE2 is part of the renin–angiotensin system and exhibits carboxypeptidase catalytic activity, breaking down Ang II into angiotensin-(1-7) [10], inducing vasodilation. Furthermore, this protein is expressed in a wide variety of tissues, such as the intestine, kidney and blood vessels, but its expression is especially high in alveolar type 2 cells in the lungs [3,11,12]. This expression profile explains the high incidence of pneumonia and bronchitis in those patients with a severe COVID-19 infection.

Once the viral infection has been detected, the human body activates the defense mechanism that comprises the initiation of an inflammatory response where reactive molecules are released, such as oxygen and nitrogen species (ROS and RNS) [13,14,15]. In non-human primates and other mammals, the expression of the high-throughput nitric oxide (NO) synthase NOS2 is one of the early responses intended to impair microbial replication and promote genomic injury [16,17,18]. Moreover, the reaction of NO with superoxide yields peroxynitrite, a short-lived radical that induces both oxidation and nitration in proteins and that is also a potent antiviral, anti-microbial and anti-parasitic molecule [19]. NO defense against viruses is not only based in oxidative damage, but also in post-translational protein modifications known as nitration and S-nitrosylation [20,21,22]. In this regard, the main mechanism for NO-mediated antiviral activity is the S-nitrosylation of cysteine, which is present in the catalytic center of many enzymes [20]. Thus, NO can target cysteine-containing proteases, ribonucleotide reductases and reverse transcriptases. Moreover, NO can also modify the zinc fingers and related domains of viral proteins as well as nucleocapsid proteins [23]. Thus, NO has been shown to limit the infection of different viruses, such as herpes simplex virus type 1 (HSV-1) [24,25], Epstein-Barr virus (EBV) [26] and coxsackievirus [27]. In the case of SARS-CoV, NO has been shown to both limit viral RNA production and reduce viral binding to its receptor ACE2 by affecting palmitoylation of the spike protein [28], thus hindering viral infection [28,29,30]. Consequently, the potential role of NO in modulating SARS-CoV-2 infection has been suggested [31,32].

Since NO is highly reactive and has a short half-life [33], most of the studies mentioned use NO donors that enable controlled NO delivery in biological systems. For the experiments described in this article, we selected two stable NO donors: DETA-NONOate as a slow-releasing, long-lasting donor [34] and S-nitrosoglutathione (GSNO) as a fast-releasing, short-lasting donor [35].

Importantly, it has been demonstrated that humans minimally use this NO-dependent pathway to fight against pathogens [36]. Several reasons have been proposed to explain the reduced expression of the NOS2 gene in humans, ranging from a different structure of the human NOS2 promoter [18] to epigenetic modifications [37], the presence of selective interfering miRNA, such as miRNA-939 [38], and polymorphic variants in a (CCTTT)n repeat present in the human NOS2 promoter [39]. For this reason, exogenous NO has already been proposed as beneficial therapy in some diseases such as neonatal respiratory distress syndrome, pulmonary hypertension or ARDS mainly due to its vasodilatory and anti-inflammatory effects [40]. In fact, NO administration has been evaluated in many different clinical trials, showing to be beneficial while safe in most cases [41]. 

Thus, taking advantage of the use of NO donors that can continuously release NO with different kinetics, we aim to evaluate the effect of this source of NO not only on the interaction between ACE2 and the viral spike protein but also on the catalytic activity of ACE2. As a result, these data could contribute to the evaluation of the therapeutic action of NO administration to COVID-19 patients.

## 2. Materials and Methods

### 2.1. Materials

The ACE2 substrate Mca-ACE2 (Mca-YVADAPK(Dnp)-OH) (#ES007) was from Bio-Techne (Minneapolis, MN, USA). The ACE2 inhibitor MLN-4760 (2(S)-(1(S)-carboxy-2-(3-(3,5-dichlorobenzyl)-3H-imidazol-4-yl)-ethylamino)-4-methylpentanoic acid) (#5.30616), recombinant human ACE2 (#SAE0064) and peroxynitrite (#516620) were from Sigma-Aldrich (St. Louis, MO, USA). DETA-NONOate (1,1-diethyl-2-hydroxy-2-nitroso-hydrazine sodium) (#ALX-430-014) and GSNO (S-nitrosoglutathione) (#ALX-420-002) were from Enzo Life Sciences. DETA-NONOate, GSNO and peroxynitrite were maintained under an anhydrous atmosphere. Reagents for electrophoresis were from Bio-Rad (Madrid, Spain). Tissue culture dishes were from Falcon (Lincoln Park, NJ, USA), and serum and culture media were from Invitrogen (Life Technologies/Thermo-Fisher, Madrid, Spain). Recombinant human spike proteins from SARS-CoV-2 were obtained from Sino Biological (Beijing, China) in the case of the Alpha variant (#40589-V08B1) and from Bio-Techne in the case of the Gamma variant (#AFG10795).

### 2.2. Isolation of Peritoneal Macrophages and Cell Line Cultures

Eight- to twelve-week-old C57Bl/6J wild-type (WT) mice were housed on a 12 h light–dark cycle at a temperature of 22 ± 2 °C with a relative humidity of 50 ± 10% and under pathogen-free conditions, with free access to food and water in the animal facility of the Instituto de Investigaciones Biomédicas (Madrid, Spain). Macrophages were isolated from the peritoneal cavity at 3 days after thioglycollate injection, as previously described [42]. Macrophages were centrifuged for 5 min at 300× *g*, cultured at 3 × 10^6^/well in 6-well Transwell inserts (Corning, Corning, NY, USA; 3460) in RPMI-1640 medium (Gibco, Madrid, Spain; 21875) supplemented with 10% FBS (Gibco, 10270106) and purified by adherence to the Transwell inserts for 4–6 h. Then, the macrophages were washed 3 times with PBS to remove non-adherent cells, and new RPMI + 10% FBS was added. 

The human hepatocellular cell line HepG2 and the epithelial alveolar A549 cell line carrying an ACE2 (A549-ACE2) transgene were a kind gift of Paloma Martín’s and Jesús Pérez-Gil’s labs, respectively. Both cell lines were maintained in DMEM with 10% FBS and 100 U/mL penicillin and streptomycin (Gibco, 15140-122). In the case of A549-ACE2 cells, the complete media also contained 100 μg/mL Normocin (InvivoGen, Toulouse, France; ant-nr-2) and 0.5 μg/mL puromycin (InvivoGen, ant-pr-1), the latter being used as selection antibiotic for ACE-2 expression.

### 2.3. Determination of NO Concentration

NO release by DETA-NONOate was determined using a NO electrode (ISO-NOP; World Precision Instruments, Stevenage, UK) at 37 °C. The NO accumulation in culture media and assays was determined by measuring the amount of nitrite using Griess reagent, as previously described [43].

### 2.4. ACE2 Activity Measurement

Firstly, recombinant human ACE2 was incubated with 1 μM DTT (Sigma-Aldrich, D-0632) for 30 min at RT to revert possible oxidations in the protein. Then, 10 ng of recombinant ACE2 protein (final concentration: 100 ng/mL) was incubated for 20 min in the presence of 2 mM DETA-NO, 2 mM GSNO, 200 μM peroxynitrite or 10 μM MLN-4760 (a specific inhibitor of ACE2) at RT. The ACE2 activity was determined using the fluorescent peptide Mca as a substrate. Mca was added at a final concentration of 10 μM in a final volume of 100 μL. The buffer used was composed of 0.3 M NaCl, 20 mM Tris-HCl and 0.1 mM ZnCl_2_. The activity measures were run per duplicate at 37 °C in a Synergy 4 plate reader (BioTek Instruments, Winooski, VT, USA) for 4 h.

### 2.5. Spike-Binding Assay by Flow Cytometry

Flow cytometry experiments were carried out in a Cytoflex S (Becton Dickinson, Madrid, Spain). Cells were trypsinized for 5 min at 37 °C and 5% CO_2_, and trypsin was neutralized with DMEM + 10% FBS. Cells were centrifuged at 420× *g* at RT for 5 min. Cells were incubated with DETA-NO or GSNO, as indicated, for 15 min at RT in Ligand Binding Buffer (LBB, 1% BSA, 0.05% sodium azide, 0.1 mM CaCl_2_). At the same time, the spike protein was placed in an Eppendorf tube and incubated in the same conditions. Cells were centrifuged again at 420× *g* for 5 min at RT. Cells were resuspended with the spike-containing solution, and the same concentration of NO donors was added again. The spike was incubated with the cells for 45 min at 4 °C with constant agitation. Another centrifugation step was performed, and for the Gamma-variant spike experiments, cells were resuspended in LBB and readily analyzed. For the Alpha-variant spike experiments, cells were resuspended in an LBB solution containing α-His AF488 antibody (Bio-Techne, IC0501G) and incubated for 30 min at 4 °C with constant agitation and in the darkness. After that, cells were centrifuged, resuspended in LBB, and analyzed. Cell viability was always determined by DAPI staining (2 μM; Invitrogen, D1306). Dead cells (DAPI^+^) were excluded from the final analysis. Experiments were analyzed using CytExpert software (Version 2.4.0.28, Beckman Coulter, Indianapolis, IN, USA). 

### 2.6. Spike Binding in A549-ACE2 Macrophage Transwell Cultures

After peritoneal macrophages had been seeded above in the Transwell inserts, 2.5 × 10^5^ A549-ACE2 cells were plated in the bottom well and maintained in RPMI + 10% FBS. One day later, the medium was replaced, and new RPMI supplemented with 10% FBS and 500 μM arginine was added in both compartments. To induce macrophage NO production, 1 μg/mL LPS (from *Salmonella enterica* serotype Typhimurium; Sigma-Aldrich, L7261) and 20 ng/mL IFNγ (Peprotech, Cranbury, NJ, USA; 315-05) were added to the upper compartment, and cells were incubated for 24 h. Then, 250 ng of spike protein was added to the lower compartment and incubated for 1 h at 4 °C with constant agitation and in the darkness. After that, medium from the upper compartment was removed and analyzed for nitrite production using the Griess reagent assay. A549-ACE2 cells in the bottom were trypsinized for 5 min at 37 °C and 5% CO_2_. Trypsinization was blocked with RPMI + 10% FBS, and cells were centrifuged at 420× *g* for 5 min at RT. Cells were resuspended in LBB and readily analyzed in the flow cytometer. DAPI staining was again used to exclude non-living cells.

### 2.7. Immunofluorescence Staining of A549-ACE2 Cells

A549-ACE2 cells were seeded at 5 × 10^4^ cells/well in 8-well culture slides (Corning, 354118) and maintained in DMEM with 10% FBS and antibiotics for 24 h at 37 °C and 5% CO_2_. Cells were treated with 2 mM DETA-NO or 2 mM GSNO for 15 min, washed with PBS and then treated again in the same way for 45 min. The medium was removed, and cells were washed with PBS. A fixation step was performed by adding 4% paraformaldehyde for 20 min at RT. Cells were blocked and permeabilized using a solution containing 3% BSA, 0.3% Triton X-100 and 2% goat serum for 1 h at RT. Another washing step was performed, and cells were incubated with ACE2 antibody (Novus Biologicals, Madrid, Spain; NBP2-67692) at a concentration of 1:100 overnight at 4 °C. Samples were incubated with the corresponding secondary antibody goat α-rabbit, which was combined with Alexa Fluor AF546 (Invitrogen, A11035), at 1:500 for 3 h. DAPI was added to stain nuclei at a 1:500 dilution for 20 min. Finally, the coverslips were mounted in ProLong Gold Antifade reagent (Invitrogen, P36934) and examined using a Zeiss LSM710 confocal microscope. Fluorescence intensity values were obtained with Image J software (Version 1.54i, Wayne Rasband (NIH), USA).

### 2.8. Protein Analysis by Western Blot

Cells were homogenized at 4 °C in a lysis buffer containing 10 mM Tris-HCl, pH 7.5, 1 mM MgCl_2_, 1 mM EGTA, 10% glycerol, 0.5% CHAPS (C3023, Sigma-Aldrich) and protease and phosphatase inhibitor cocktails (P8340, P5726, P0044, Sigma-Aldrich). Samples were vortexed for 30 min and centrifuged at 12,000× *g* for 15 min at 4 °C. Supernatants were stored at −20 °C. Protein concentration was determined by the Bradford assay (Bio-Rad). Equal amounts of protein (40 μg) from each fraction obtained were loaded into 8% SDS-PAGE. Proteins were size-fractionated, transferred to a PVDF membrane (Bio-Rad, 1704157) and, after blocking with 5% nonfat milk, incubated overnight with ACE2 antibody (Novus Biologicals, NBP2-67692) and β-actin antibody (Sigma-Aldrich, A-5441) as a loading control. Blots were developed by the ECL protocol, and different exposition times were performed for each blot to ensure the linearity of the band intensities. Values of densitometry were determined using Image J software. 

### 2.9. Statistical Analysis

The values in graphs correspond to mean ± SD. The statistical significance of differences between the means was determined with GraphPad Prism 9.0.0. (GraphPad Software) using a one-way analysis of variance (ANOVA) followed by a Bonferroni post hoc test or Student’s *t*-test, as appropriate. A *p*-value < 0.05 was considered to be significant.

## 3. Results

### 3.1. ACE2 Activity Is Inhibited by NO Donors and Peroxynitrite

Although NO is famously known to inhibit ACE [44], its effects on ACE2 are still unknown. Thus, we aimed to evaluate the effects of NO and its byproduct peroxynitrite on ACE2 activity. Considering the reduced capacity of humans to express NOS2 (the enzyme responsible for the high-throughput NO synthesis) in response to pro-inflammatory challenges [16,17,18,38,39], we opted for strategies based on the use of NO donors, such as the short-lived NO donor S-nitrosoglutathione (GSNO) [45] and the long-lived NO donor DETA-NO [34]. We also used peroxynitrite, a highly reactive agent that can induce oxidation and nitration in proteins [46,47,48]. 

In the case of DETA-NO, the release of NO was extended at a continuous rate for at least 24 h. As Figure 1A shows, the rate of release of 1 mM DETA-NO in DMEM at pH 7.4 accounts for a concentration of 0.52 ± 0.07 μM, which is in the range of the activity of NOS2 expression in murine peritoneal macrophages activated under pro-inflammatory conditions [49]. After that, we explored the effect of this NO release on the enzymatic activity of ACE2, using the recombinant human protein and a synthetic peptide substrate linked to 7-methoxycoumarin-4-acetate (Mca), which becomes fluorescent when cleaved. Both DETA-NO and peroxynitrite caused a mild inhibition, while GSNO strongly inhibited ACE2 activity, almost at the same rate as that of the specific ACE2 inhibitor MLN-4760 (Figure 1B). This inhibition was statistically significant for all the agents used when looking at both the peptide fluorescence (Figure 1C) and the enzymatic maximal activity (Figure 1D) at 20 min. Consistently, the inhibition persisted throughout 90 min and at least until 4 h after the beginning of the analysis (Appendix A) Thus, both NO donors and peroxynitrite can limit ACE2 enzyme activity, with this inhibition being greater when the fast NO-releasing donor GSNO was used. 

### 3.2. DETA-NO Reduces the Binding of the Alpha Variant of the SARS-CoV-2 Spike Protein to ACE2 in HepG2 Cells

The inhibition of ACE2 activity by NO donors prompted us to investigate if these molecules can interfere in the interaction between the spike protein of SARS-CoV-2 and the ACE2 receptor expressed in mammalian cells. To achieve this, we selected the human hepatocellular cell line HepG2, as it showed ACE2 protein expression by Western Blot analysis (Appendix A). Moreover, we used the Alpha variant of the spike, as we wanted to test the effects of NO in the original variant before the different mutations enhanced viral infectivity.

For the binding experiments, we exposed HepG2 cells and spike to DETA-NO to let the pertinent modifications take place. Then, cells were incubated with the spike, and the spike proteins that were bound to human cells were quantified by flow cytometry analysis. As Figure 2A shows, the viability of the cell line HepG2 was not affected by treatment with DETA-NO or the recombinant viral spike protein. Interestingly, the binding of the Alpha variant of the spike protein to HepG2 cells was reduced after incubation with DETA-NO in a dose-dependent manner (Figure 2B,C), confirming our hypothesis that NO donors can inhibit the spike–ACE2 interaction.

### 3.3. NO Donors Inhibit the Binding of the Gamma Variant of the SARS-CoV-2 Spike Protein in ACE2-Transfected A549 Cells

After showing the inhibition using the Alpha variant of the spike, we wondered whether this effect was consistent in a different, more actual variant, as the spike protein is known to accumulate a great part of the mutations that SARS-CoV-2 suffers to enhance its infectivity. For this reason, for these experiments, we used the Gamma variant of the spike protein. Furthermore, we wanted to test the inhibition in a lung model, which is the main route of entrance for SARS-CoV-2. Hence, we selected the alveolar epithelial cell line A549. However, as it did not show ACE2 expression (Appendix A), we used the same cell line but transfected with ACE2 (A549-ACE2). As expected, this cell line showed expression of ACE2, which was much higher than that of HepG2 cells (Appendix A). These ACE2-transfected cells offer the possibility to titrate the binding of the spike due to the high expression of ACE2. Accordingly, incubation with increasing concentrations of the spike in the presence or absence of DETA-NO did not affect cell viability (Figure 3A). 

As Figure 3B–D shows, DETA-NO significantly reduced the binding of the spike at concentrations of 125 ng and 500 ng, although some inhibition can be observed at all concentrations. Furthermore, we investigated the effect of the short-lasting NO donor GSNO in this cell model. Again, no differences in cell viability were observed (Appendix A), and there was a significant inhibition of the spike binding when 2 mM GSNO was added (Appendix A). To ensure that the effect was a consequence of the inhibition of the binding, we treated A549-ACE2 cells with DETA-NO or GSNO and looked at ACE2 expression in immunofluorescence (Appendix A). None of the treatments modified the distribution of ACE2 (Appendix A) or its expression levels (Appendix A). Importantly, our results show that both NO donors inhibited the interaction between ACE2 and the viral spike protein, while being harmless to cell viability.

### 3.4. Murine Peritoneal Macrophages Treated with LPS and IFNγ Inhibit the Binding of the Spike Protein to A549-ACE2 Cells

To also determine the effect of endogenous NO release on the binding of the recombinant spike to the cells, we used primary cultures of murine peritoneal macrophages treated in the absence or presence of LPS and IFNγ to induce NOS2 expression and activity [50]. As Figure 4A shows, neither the treatment of macrophages with the spike nor the treatment with the pro-inflammatory stimuli affected the viability of A549-ACE2 cells. The synthesis and release of NO by pro-inflammatory macrophages were assessed by measuring the accumulation of nitrite in the culture medium. As Figure 4B shows, treatment with the spike did not induce the synthesis of NO, while stimulation with LPS and IFNγ for 24 h induced a 10-fold increase in nitrite accumulation. Interestingly, the binding of the spike to A549-ACE2 cells was significantly reduced in the co-culture of NO-releasing murine macrophages (Figure 4C,D). These results confirm that the release of NO from a living source, the peritoneal macrophages, can also impair binding of the spike to ACE2.

## 4. Discussion

One of the main reasons for the success of SARS-CoV-2 infection is its great efficiency in binding the human enzyme ACE2 through the viral protein termed spike [51]. Once the spike recognizes and binds to ACE2, the virus enters the cell [4], where it uses the replication machinery to thrive [52]. For this reason, researchers have invested much effort in developing different strategies to inhibit the spike–ACE2 interaction [53,54]. 

Nitric oxide (NO) is a gaseous mediator that affects different biological mechanisms, inducing post-translational modifications in proteins, regulating the vascular tone or helping in the fight against pathogens [55]. Therefore, NO has been considered as a therapeutic option for different diseases. In fact, its use is approved for pulmonary hypertension in neonates [56] and for ARDS [57] and pneumonia [58] in adults. In the case of COVID-19, a great number of clinical trials have been developed for NO use [41]. For example, NO was helpful in mild [59] COVID-19 patients and patients with viral pneumonia [60] and accelerated SARS-CoV-2 viral clearance [61]. Thus, we considered evaluating whether this beneficial action can be associated with a lesser infection capacity mediated by a reduced binding capacity of the viral spike to ACE2, which could be induced by the presence of NO.

Many authors have suggested that NO could be beneficial in COVID-19 disease [41,62,63] not only because of its effects on the immune system and vasculature [64] but also due to its potential as an antiviral agent [65]. These effects are mainly mediated by IFNγ, which stimulates the production of NO through NOS2 induction [24] in other species, as this pathway is lacking in humans. However, there is controversy about this last claim, as some authors believe that human macrophages are able to express NOS2 [66,67], while other authors do not [68,69]. Therefore, we decided to use NO donors considering that NO is short-lived [33], which is why a constant generation of NO is needed to be able to study its actions. This long-term generation of NO is reached by using NO donors.

Therefore, we first addressed the effects of NO in ACE2 activity, contemplating how they would affect diseases like COVID-19. Our results show that NO from two different sources was able to reduce ACE2 activity. While DETA-NO, a long-lasting and slow-releasing NO donor caused mild inhibition, the short-lasting, fast-releasing GSNO blocked ACE2 activity at a similar scale to that of the specific inhibitor MLN-4760. This difference could be because GSNO reaches a higher concentration of NO at the timing used, as it liberates NO faster than DETA-NO. Interestingly, although GSNO has a shorter half-life, the inhibition was maintained throughout the experiment, which lasted for 4 h, indicating that GSNO is stable for at least that lapse of time. Longer-lasting experiments are needed to discern whether this inhibition is reversible. Moreover, peroxynitrite also partially blocked ACE2 activity, which opens the door to a role of this molecule in the renin–angiotensin–aldosterone system (RAAS), a possibility that has not been studied so far.

Endogenously, the angiotensin-converting enzyme (ACE) plays an important role in the RAAS by breaking down Ang I into Ang II, which binds to angiotensin II type 1 receptors (AT1R), inducing vasoconstriction. The relationship of ACE with NO has been studied in depth. It is widely known that NO inhibits ACE [44] and downregulates AT1R [70], inducing vasodilation. At the same time, ACE inhibitors enhance the levels of NO produced by the endothelial form of NOS (eNOS) [31,71]. Thus, our results indicating that NO inhibits ACE2 activity are somewhat surprising. These results could mean that NO is stimulating a negative loop for its own production through inhibiting the ACE2–Ang-(1-7)–Mas1 axis, as this signaling pathway is known to induce NO production [72]. Although the inhibition of ACE2 may suggest that NO blunts ACE2-mediated vasodilating properties, it is clear that this is not the case, provided that NO is broadly known for its vasodilating properties. One explanation is that, as NO also blocks ACE activity, it is unlikely that Ang II accumulates in this context, while its detrimental actions would be hindered by NO-mediated AT1R downregulation. Moreover, Ang-(1-7) levels could be maintained through another pathway, which implies Ang I processing by neprilysin [73]. However, more experiments are needed in order to discern this possibility. 

Interestingly, some authors have proposed that excessive activation of ACE2 in circulation produced by SARS-CoV-2 could be detrimental [71]. This hypothesis is based on the fact that ACE2 activity is generally increased in COVID-19 patients and positively correlated to pathology prognosis [74]. Therefore, overactivation of ACE2 could induce a compensatory increase in the ACE–Ang II–AT1R axis, ultimately leading to increased inflammation and blood pressure [75,76,77]. In fact, the SARS-CoV-2 spike itself was shown to increase recombinant ACE2 activity [78]. Considering this additional adverse effect of spike binding to ACE2, NO inhibition of both ACE and ACE2 would prevent this compensatory response, abolishing the pro-inflammatory and vasoconstrictive effects of the ACE–Ang II–AT1R axis. 

After that, we focused on the effect of NO donors on the binding of the spike protein. In our study, we have shown that NO inhibits the SARS-CoV-2 spike binding to its receptor ACE2 in a cell line that endogenously expresses ACE2 (HepG2), showing that this impairment would happen in an actual human being. We also observed the inhibition in an epithelial lung cell line (A549), one of the main infection routes of the virus. A combination of both models demonstrates that NO can inhibit viral entry into ACE2-expressing cells in the human lungs, which could hinder SARS-CoV-2 infection and replication. Furthermore, we used Alpha and Gamma variants of the spike protein, showing similar results. Thus, this inhibition seems to be resistant to viral mutations, although the same study should be replicated in novel strains that have recently appeared. These results agree with previous reports where NO was shown to inhibit SARS-CoV-2 viral entry through ACE2 S-nitrosylation [79] or spike nitration [80].

The most-used therapeutic approaches for COVID-19 include small molecules (i.e., chloroquine, suramin, Paxlovid, remdesivir, etc.) [81] and specific antibodies that prevent the spike–ACE2 interaction [82]. However, although the search for small molecules has reduced their relevance in the field, new developments in the design of antibodies have become the therapeutic option of choice. Nevertheless, therapies are broadly categorized as those targeting the viral infection and those targeting the host response. Interestingly, NO would play both roles, inhibiting the viral entry while ameliorating the host inflammatory state, thrombosis and ARDS. Thus, as NO has been shown to be safe and effective against other pathologies and in COVID-19 clinical trials, it represents a promising therapeutic strategy that is worthy of further investigation.

Nevertheless, the translation of NO into the clinic still poses a challenge, as NO donors or carriers exhibit short half-lives and can be toxic, and it is hard to control NO release [55]. Thus, inhaled NO is probably the best option, as it is safe and effective [83]. 

Finally, while the present study shows that NO could play a role in preventing SARS-CoV-2 infection, it may also have therapeutic potential for other viral diseases that rely on similar receptor-mediated entry mechanisms, as is common for most of the coronaviruses [84].

## 5. Conclusions

In conclusion, we have shown that NO impairs the binding of the spike protein to human cells, potentially preventing SARS-CoV-2 infection by inhibiting its entry into the cell. Importantly, these effects were observed across different experimental models. Additionally, NO decreased ACE2 activity, which could counterbalance the spike-mediated increase and the concomitant compensatory response. These results support the hypothesis that NO can serve as therapeutic agent in COVID-19 due to its antiviral effects, which add to its already known role as an immune response modulator and lung vasodilator. However, further research is needed to elucidate the specific molecular mechanism for the binding impairment and the consequences of ACE2 inhibition in presence of SARS-CoV-2. In this sense, because most animal models express NOS2 in response to infectious challenges, the use of mice lacking the *NOS2* gene or the administration of selective inhibitors of NOS2 activity may help provide preclinical data on the antiviral actions of NO in this context. 

## Figures and Tables

**Figure 1 antioxidants-13-01301-f001:**
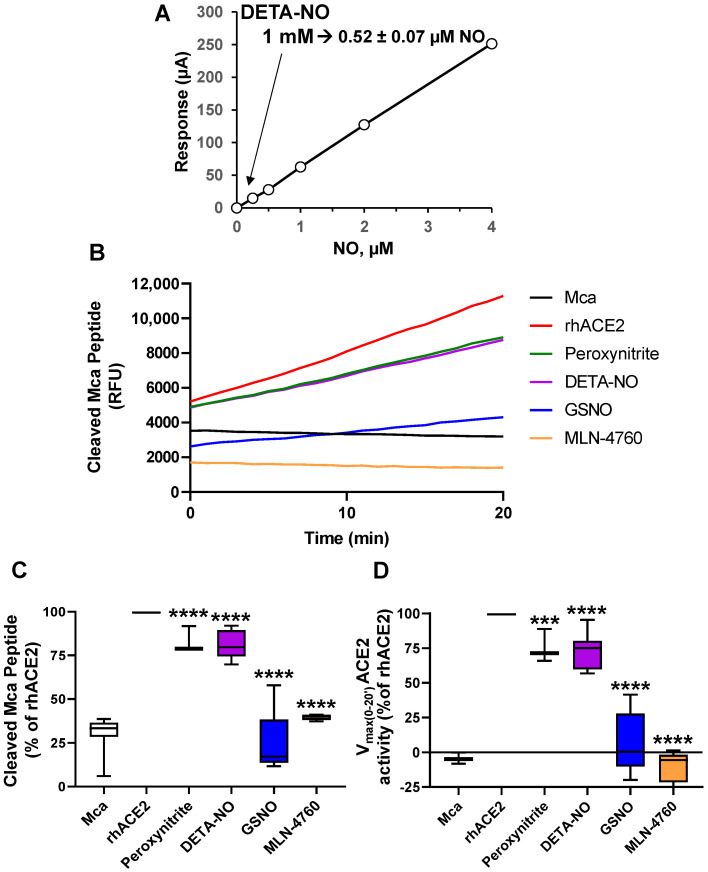
NO donors and peroxynitrite inhibit the activity of recombinant human ACE2. (**A**) The release of NO from a solution of 1 mM DETA-NO in HBSS at pH 7.4 was determined using a selective NO electrode. (**B**) ACE2 activity was determined using 10 ng of the recombinant human ACE2 enzyme (rhACE2, 100 ng/mL) and 10 μM of the Mca-ACE2 substrate that releases the fluorescent methyl coumarin. The NO donors were added at 2 mM GSNO, 2 mM DETA-NO and 0.2 mM ONOO^−^ (peroxynitrite). The ACE2 inhibitor MLN-4760 was used at 10 μM. A condition was included with Mca alone (without rhACE2) as a negative control of the enzymatic reaction. The figure shows a representative experiment. (**C**) The fluorescence of the cleaved Mca peptide after 20 min of each treatment is shown. (**D**) The enzymatic maximum velocity of ACE2 between minutes 0 and 20 from the start of the reaction is shown. The results show the mean ± S.D. of four distinct experiments (**C**,**D**); *** *p* < 0.001, **** *p* < 0.0001 vs. Mca.

**Figure 2 antioxidants-13-01301-f002:**
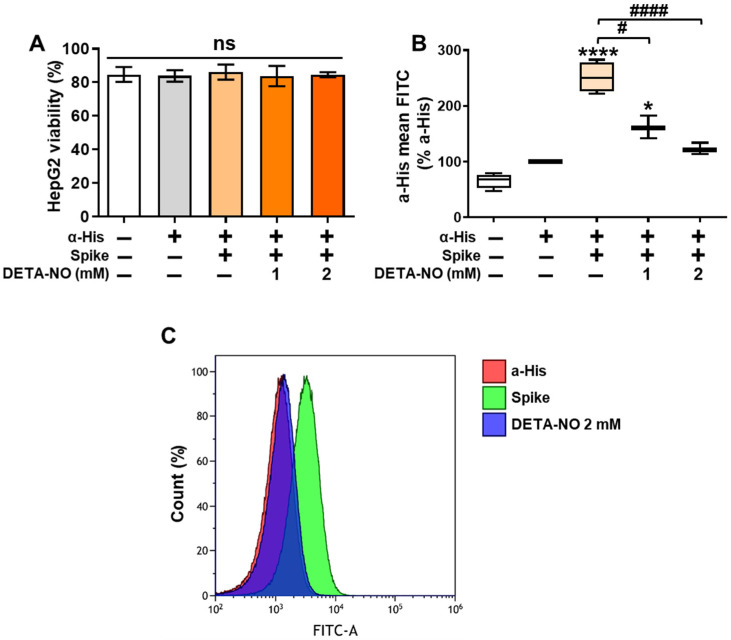
DETA-NO inhibits the binding of a recombinant Alpha variant of the SARS-CoV-2 spike to HepG2 cells. (**A**) Cells (5 × 10^5^) were incubated with 1 or 2 mM DETA-NO for 15 min, followed by incubation with 1 μg spike and the same concentrations of DETA-NO for another 45 min. The viability was determined with DAPI staining. (**B**) The binding of the Alpha variant of the spike to cells was determined by flow cytometry using a fluorescent α-His antibody targeted to a His motif present in the spike protein. (**C**) Representative histogram plots of the fluorescence of the spike protein bound to the HepG2 cells. The results show the means ± S.D. from 4 different assays (**A**,**B**); * *p* < 0.05, **** *p* < 0.0001 vs. the untreated control condition; # *p* < 0.05, #### *p* < 0.0001 vs. the spike condition. ns: not significant.

**Figure 3 antioxidants-13-01301-f003:**
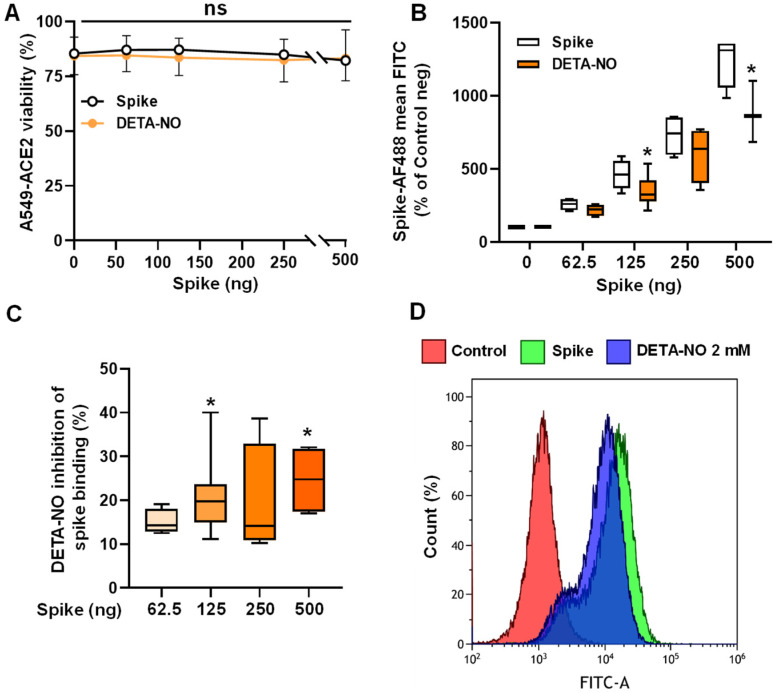
DETA-NO inhibits the binding of the recombinant Gamma variant of SARS-CoV-2 spike to A549-ACE2 cells. (**A**) Cells (5 × 10^5^) were incubated with 2 mM DETA-NO for 15 min, followed by incubation with the indicated concentrations of spike and DETA-NO for another 45 min. The viability was determined by DAPI staining. (**B**,**C**) The dose-dependent binding of the Gamma variant of the spike to A549-ACE2 cells was determined by flow cytometry. The results are shown as mean FITC (**B**) or percentage of binding inhibition (**C**). (**D**) Representative histogram plots of the fluorescence of the spike protein bound to the A549-ACE2 cells. The results show the mean ± S.D. of 4 independent experiments (**B**,**C**); * *p* < 0.05 vs. the control condition. ns: not significant.

**Figure 4 antioxidants-13-01301-f004:**
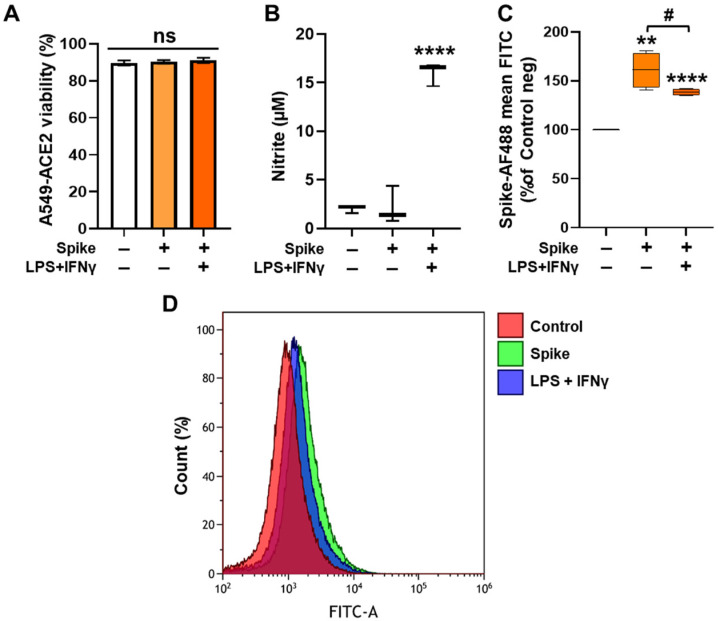
Treatment of murine peritoneal macrophages with LPS and IFNγ inhibits the binding of the recombinant SARS-CoV-2 spike to A549-ACE2 cells. (**A**) Peritoneal macrophages (3 × 10^6^) were co-cultured with A549-ACE2 cells (2.5 × 10^5^) in DMEM medium with FBS and incubated for 24 h with 1 μg/mL LPS and 20 ng/mL IFNγ. Then, 125 ng of the spike was added for 45 min, and the A549-ACE2 cell viability was determined with DAPI staining. (**B**) The accumulation of nitrite in the culture medium was determined with Griess reagent. (**C**) The binding of the spike protein to A549-ACE2 cells was determined by flow cytometry. (**D**) Representative histogram plots of the fluorescence of the spike protein bound to the A549-ACE2 cells. The results show the mean ± S.D. from 3 different assays (**A**–**C**); ** *p* < 0.01; **** *p* < 0.0001 vs. the untreated control condition; # *p* < 0.05 vs. the spike condition. ns: not significant.

## Data Availability

Data are contained within the article.

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
