# Peer review of "Potential Beneficial Role of Nitric Oxide in SARS-CoV-2 Infection: Beyond Spike-Binding Inhibition"

_antioxidants, 2024, doi:10.3390/antiox13111301_

Round 1
Reviewer 1 Report
The manuscript explores the role of nitric oxide (NO) in inhibiting the interaction between the SARS-CoV-2 spike protein and the human ACE2 receptor. The study demonstrates that NO donors, such as DETA-NONOate and GSNO, reduce ACE2 activity and decrease spike protein binding to ACE2 in human cells. The findings are consistent across different SARS-CoV-2 variants, suggesting that NO could serve as a potential therapeutic agent against COVID-19. The introduction provides a detailed background on SARS-CoV-2 and its infection mechanisms, but it may contain information that is too basic or widely known by the target scientific audience. Simplifying this section by focusing more on the specific knowledge gaps related to NO’s antiviral potential could enhance readability. For instance, a brief summary of SARS-CoV-2 could suffice, while placing more emphasis on the novel aspects of NO’s interaction with the virus and its potential therapeutic use. Highlighting the study's objective to fill these gaps early in the introduction would also improve the narrative flow.
While the introduction touches on NO’s antiviral properties, it lacks a clear link between how NO has been shown to inhibit other viruses, like HSV and EBV, and how these mechanisms might extend to SARS-CoV-2. Explicitly making this connection would help justify the relevance of exploring NO in this context. You could include a sentence explaining that NO’s mechanisms against other viruses might similarly be effective against SARS-CoV-2, making it clearer why this study is worth pursuing.
Additionally, the introduction could benefit from more recent findings on ACE2 and NO interactions, as much of the referenced literature is from the early stages of the pandemic. It would be useful to incorporate newer studies from the past two years, providing updated insights on NO’s role in COVID-19. This will help the paper stay current and show that the research builds upon the latest developments in the field. When introducing NO donors like DETA-NONOate and GSNO, it would be clearer to discuss their relevance earlier in the introduction, particularly when describing NO’s antiviral potential. Explaining the choice of these compounds up front will help guide readers through the experimental rationale more smoothly.
In the discussion, it could be valuable to compare NO's potential therapeutic effects to other treatments currently used for COVID-19, such as antivirals or monoclonal antibodies. This would highlight the unique advantages NO might offer or potential limitations compared to these existing therapies. Adding a comparison could strengthen the discussion by placing NO in the context of the broader treatment landscape for COVID-19. For example, NO’s mechanism of action could be contrasted with drugs like remdesivir or Paxlovid, and any potential for synergistic effects could be explored.
The study’s limitations, particularly the fact that the experiments were conducted in vitro, are not fully addressed. Acknowledging these limitations and suggesting directions for future research would provide a more balanced discussion. For instance, a paragraph highlighting the need for in vivo studies or clinical trials would improve the paper’s credibility. It would also be beneficial to discuss potential challenges in translating these findings to clinical settings, such as optimizing NO dosage or delivery methods.
The discussion briefly mentions that NO inhibits ACE2, but it would be useful to delve deeper into the physiological consequences of this inhibition, especially considering the complex roles ACE2 plays in COVID-19 patients. ACE2 is not only involved in viral entry but also has protective effects, particularly in the cardiovascular and respiratory systems. Discussing the broader implications of inhibiting ACE2 activity in the context of these systems could help readers understand the potential risks and benefits of NO-based treatments more fully.
Finally, the therapeutic potential of NO in COVID-19 management could be explored in more depth. The discussion would benefit from a more thorough analysis of how NO therapy could be integrated into existing treatment protocols. This could include considerations such as patient populations (e.g., mild vs. severe cases), specific dosages, and methods of administration (e.g., inhalation). Addressing possible safety concerns or contraindications could also be valuable for clinicians considering NO as a treatment option. In conclusion, the paper could be strengthened by providing a more assertive summary of its broader implications. While the study shows that NO could play a role in preventing SARS-CoV-2 infection, it may also have therapeutic potential for other viral diseases that rely on similar receptor-mediated entry mechanisms. Ending with a statement about NO’s broader relevance in antiviral therapies could leave a stronger impact.
The manuscript explores the role of nitric oxide (NO) in inhibiting the interaction between the SARS-CoV-2 spike protein and the human ACE2 receptor. The study demonstrates that NO donors, such as DETA-NONOate and GSNO, reduce ACE2 activity and decrease spike protein binding to ACE2 in human cells. The findings are consistent across different SARS-CoV-2 variants, suggesting that NO could serve as a potential therapeutic agent against COVID-19. The introduction provides a detailed background on SARS-CoV-2 and its infection mechanisms, but it may contain information that is too basic or widely known by the target scientific audience. Simplifying this section by focusing more on the specific knowledge gaps related to NO’s antiviral potential could enhance readability. For instance, a brief summary of SARS-CoV-2 could suffice, while placing more emphasis on the novel aspects of NO’s interaction with the virus and its potential therapeutic use. Highlighting the study's objective to fill these gaps early in the introduction would also improve the narrative flow.
While the introduction touches on NO’s antiviral properties, it lacks a clear link between how NO has been shown to inhibit other viruses, like HSV and EBV, and how these mechanisms might extend to SARS-CoV-2. Explicitly making this connection would help justify the relevance of exploring NO in this context. You could include a sentence explaining that NO’s mechanisms against other viruses might similarly be effective against SARS-CoV-2, making it clearer why this study is worth pursuing.
Additionally, the introduction could benefit from more recent findings on ACE2 and NO interactions, as much of the referenced literature is from the early stages of the pandemic. It would be useful to incorporate newer studies from the past two years, providing updated insights on NO’s role in COVID-19. This will help the paper stay current and show that the research builds upon the latest developments in the field. When introducing NO donors like DETA-NONOate and GSNO, it would be clearer to discuss their relevance earlier in the introduction, particularly when describing NO’s antiviral potential. Explaining the choice of these compounds up front will help guide readers through the experimental rationale more smoothly.
In the discussion, it could be valuable to compare NO's potential therapeutic effects to other treatments currently used for COVID-19, such as antivirals or monoclonal antibodies. This would highlight the unique advantages NO might offer or potential limitations compared to these existing therapies. Adding a comparison could strengthen the discussion by placing NO in the context of the broader treatment landscape for COVID-19. For example, NO’s mechanism of action could be contrasted with drugs like remdesivir or Paxlovid, and any potential for synergistic effects could be explored.
The study’s limitations, particularly the fact that the experiments were conducted in vitro, are not fully addressed. Acknowledging these limitations and suggesting directions for future research would provide a more balanced discussion. For instance, a paragraph highlighting the need for in vivo studies or clinical trials would improve the paper’s credibility. It would also be beneficial to discuss potential challenges in translating these findings to clinical settings, such as optimizing NO dosage or delivery methods.
The discussion briefly mentions that NO inhibits ACE2, but it would be useful to delve deeper into the physiological consequences of this inhibition, especially considering the complex roles ACE2 plays in COVID-19 patients. ACE2 is not only involved in viral entry but also has protective effects, particularly in the cardiovascular and respiratory systems. Discussing the broader implications of inhibiting ACE2 activity in the context of these systems could help readers understand the potential risks and benefits of NO-based treatments more fully.
Finally, the therapeutic potential of NO in COVID-19 management could be explored in more depth. The discussion would benefit from a more thorough analysis of how NO therapy could be integrated into existing treatment protocols. This could include considerations such as patient populations (e.g., mild vs. severe cases), specific dosages, and methods of administration (e.g., inhalation). Addressing possible safety concerns or contraindications could also be valuable for clinicians considering NO as a treatment option. In conclusion, the paper could be strengthened by providing a more assertive summary of its broader implications. While the study shows that NO could play a role in preventing SARS-CoV-2 infection, it may also have therapeutic potential for other viral diseases that rely on similar receptor-mediated entry mechanisms. Ending with a statement about NO’s broader relevance in antiviral therapies could leave a stronger impact.
Author Response
The manuscript explores the role of nitric oxide (NO) in inhibiting the interaction between the SARS-CoV-2 spike protein and the human ACE2 receptor. The study demonstrates that NO donors, such as DETA-NONOate and GSNO, reduce ACE2 activity and decrease spike protein binding to ACE2 in human cells. The findings are consistent across different SARS-CoV-2 variants, suggesting that NO could serve as a potential therapeutic agent against COVID-19. The introduction provides a detailed background on SARS-CoV-2 and its infection mechanisms, but it may contain information that is too basic or widely known by the target scientific audience. Simplifying this section by focusing more on the specific knowledge gaps related to NO’s antiviral potential could enhance readability. For instance, a brief summary of SARS-CoV-2 could suffice, while placing more emphasis on the novel aspects of NO’s interaction with the virus and its potential therapeutic use. Highlighting the study's objective to fill these gaps early in the introduction would also improve the narrative flow.
We thank the reviewer for this synthesis of our work. Following the suggestions, we have reduced the scope of information regarding COVID-19 pathogenesis and have made it more concise. Moreover, we have underscored the known actions of NO in COVID-19.
While the introduction touches on NO’s antiviral properties, it lacks a clear link between how NO has been shown to inhibit other viruses, like HSV and EBV, and how these mechanisms might extend to SARS-CoV-2. Explicitly making this connection would help justify the relevance of exploring NO in this context. You could include a sentence explaining that NO’s mechanisms against other viruses might similarly be effective against SARS-CoV-2, making it clearer why this study is worth pursuing.
This section has been expanded, according to the recommendations provided. The anti-viral mechanisms of NO are multiple and probably depend not only on the direct action of NO on the virus (for example, the inhibition of coxsackievirus B3 proteinases, which reduces the cardiotoxicity of the infection), but also on the specific cell receptors that mediate entry into the cell. We think that this section has been improved following the comments of the reviewer.
Additionally, the introduction could benefit from more recent findings on ACE2 and NO interactions, as much of the referenced literature is from the early stages of the pandemic. It would be useful to incorporate newer studies from the past two years, providing updated insights on NO’s role in COVID-19. This will help the paper stay current and show that the research builds upon the latest developments in the field. When introducing NO donors like DETA-NONOate and GSNO, it would be clearer to discuss their relevance earlier in the introduction, particularly when describing NO’s antiviral potential. Explaining the choice of these compounds up front will help guide readers through the experimental rationale more smoothly.
Thank you for these comments that we have considered in the Introduction section. We updated the references on the proposals for the use of NO in COVID-19 in the discussion section, which is where this topic is covered (REF: 31 and 72). Additionally, we have included the reason for the choice of the NO donors.
In the discussion, it could be valuable to compare NO's potential therapeutic effects to other treatments currently used for COVID-19, such as antivirals or monoclonal antibodies. This would highlight the unique advantages NO might offer or potential limitations compared to these existing therapies. Adding a comparison could strengthen the discussion by placing NO in the context of the broader treatment landscape for COVID-19. For example, NO’s mechanism of action could be contrasted with drugs like remdesivir or Paxlovid, and any potential for synergistic effects could be explored.
We agree that this field is relevant in the context of NO treatment. We have included details about small molecule and antibody treatments of SARS-CoV-2 infection. In addition, as now included in the manuscript, we underscore the dual role of NO in the viral infection and the host response, which may confer advantages when compared to the other therapeutic approaches, which only target one side of the pathology.
The study’s limitations, particularly the fact that the experiments were conducted in vitro, are not fully addressed. Acknowledging these limitations and suggesting directions for future research would provide a more balanced discussion. For instance, a paragraph highlighting the need for in vivo studies or clinical trials would improve the paper’s credibility. It would also be beneficial to discuss potential challenges in translating these findings to clinical settings, such as optimizing NO dosage or delivery methods.
This is a relevant topic and we combined experiments with recombinant proteins, which provide an unbiased analysis, with cell-based assays, which confirmed the in vitro results. We recognize the importance of moving to more preclinical studies; however, considering that animal models rapidly express NOS2 in response to infectious challenges, the best approach would be the use of mice lacking NOS2. This model could help address the analysis of exogenous NO administration. Following your suggestion, these comments have been incorporated in the Conclusion section.
The discussion briefly mentions that NO inhibits ACE2, but it would be useful to delve deeper into the physiological consequences of this inhibition, especially considering the complex roles ACE2 plays in COVID-19 patients. ACE2 is not only involved in viral entry but also has protective effects, particularly in the cardiovascular and respiratory systems. Discussing the broader implications of inhibiting ACE2 activity in the context of these systems could help readers understand the potential risks and benefits of NO-based treatments more fully.
We revised this part of the discussion and added comments regarding the effects of NO on the RAAS components, including ACE, ACE2, Ang II, Ang-(1-7) and AT1R. It is our hope that the known effect of NO described in the literature and our hypothesis regarding its role in COVID-19 are now more clearly understood.
Finally, the therapeutic potential of NO in COVID-19 management could be explored in more depth. The discussion would benefit from a more thorough analysis of how NO therapy could be integrated into existing treatment protocols. This could include considerations such as patient populations (e.g., mild vs. severe cases), specific dosages, and methods of administration (e.g., inhalation). Addressing possible safety concerns or contraindications could also be valuable for clinicians considering NO as a treatment option. In conclusion, the paper could be strengthened by providing a more assertive summary of its broader implications. While the study shows that NO could play a role in preventing SARS-CoV-2 infection, it may also have therapeutic potential for other viral diseases that rely on similar receptor-mediated entry mechanisms. Ending with a statement about NO’s broader relevance in antiviral therapies could leave a stronger impact.
Following your comment, we have implemented the opinion on the role of NO in COVID-19. Since its use by inhalation has been clinically approved, we think this way is the best option for preclinical or early phases of clinical trials aimed at controlling viremias. Thus, we have included a paragraph in this regard at the end of the discussion section.
Regarding the proposed assertive phrase, we appreciate this nuanced summary that we elaborated a little on, but essentially maintaining your structure.
Reviewer 2 Report
Clinical data are beginning to emerge that NO may be useful in the treatment of COVID, but the data strongly indicate that NO not only lowers elevated blood pressure, but also inhibits the binding of Ace2 to spike, i.e. has direct antiviral activity. The efficacy of some NO-producing substances was also demonstrated here. This is a very important article that will make a huge contribution to future therapies.
I have one request regarding the way the data was summarised.
All bar plots: If there are only four data, it would be easier to see the differences if they were plotted instead of bars.
However, if these are cytometry results, using only four of their averages is a rather wasteful situation in terms of number of data. It is strongly recommended to use those histograms, which summarise the four experiments, to create a violin plot, if possible. Or a box plot is also acceptable. The reliability of the data is ultimately backed up by the number of measurements. Cytometry is a very good method in this respect, which is why Fig2C and 3D are so convincing.
a minor point:
Fig 2
"Results show the means ± S.D. from 4 different assays. *p<0.05, ****p<0.0001 vs. the untreated control condition; #p<0.05, p<0.0001 vs. the spike condition. ns: not significant. "
It is not likely that this explains C. Are not in the wrong place?
Author Response
Clinical data are beginning to emerge that NO may be useful in the treatment of COVID, but the data strongly indicate that NO not only lowers elevated blood pressure, but also inhibits the binding of Ace2 to spike, i.e. has direct antiviral activity. The efficacy of some NO-producing substances was also demonstrated here. This is a very important article that will make a huge contribution to future therapies.
We thank the reviewer for the positive comments on our work.
I have one request regarding the way the data was summarised.
All bar plots: If there are only four data, it would be easier to see the differences if they were plotted instead of bars.
However, if these are cytometry results, using only four of their averages is a rather wasteful situation in terms of number of data. It is strongly recommended to use those histograms, which summarise the four experiments, to create a violin plot, if possible. Or a box plot is also acceptable. The reliability of the data is ultimately backed up by the number of measurements. Cytometry is a very good method in this respect, which is why Fig2C and 3D are so convincing.
Following your comments, we have modified the way the results are presented. In our opinion, the actual presentation is more visual and provides the reader with cumulative information about the biological effects. We also tried using the violin representations in the Prisma software, but the visualization was less clear.
Regarding the nature of the data, those referring to ACE2 activity have been determined by fluorescence recording and quantification of substrate degradation. For the binding between ACE2 and the spike proteins we have used flow cytometry.
Detail comments
a minor point:
Fig 2
"Results show the means ± S.D. from 4 different assays. *p<0.05, ****p<0.0001 vs. the untreated control condition; #p<0.05, p<0.0001 vs. the spike condition. ns: not significant. "
It is not likely that this explains C. Are not in the wrong place?
We thank the reviewer for this observation. Measures have been taken to better specify the data to which the statistical criteria are applied.